# Confucianism and Gender Ratios of Suicide in the World: A WHO Data-Based Study

**DOI:** 10.3390/ijerph20032188

**Published:** 2023-01-25

**Authors:** Wei Wang, Jie Zhang, Wayne L. Thompson

**Affiliations:** 1Department of Sociology and Criminal Justice, University of Delaware, Newark, DE 19716, USA; 2School of Public Health, Shandong University, Jinan 250012, China; 3Department of Sociology, State University of New York at Buffalo State, Buffalo, NY 14222, USA; 4Department of Sociology, Carthage College, Kenosha, WI 53140, USA

**Keywords:** suicide, gender ratio, Confucianism, culture, China

## Abstract

This study explores how Confucianism affects suicide rates by gender. Data for the study come from the World Health Organization document “Suicide Worldwide in 2019”, which provides frequency and gender ratios for suicide rates in 183 member countries. One-way ANOVA and multiple linear regression analysis were used to examine potential differences in suicide rates and male to female ratio of suicides. Independent variables include region, income level, culture, and Confucian values that may be related to suicide. Suicide rates for Confucian countries do not show significant differences from European countries. However, these countries have lower suicide gender ratios.

## 1. Introduction

Suicide is a serious global public health issue, among the leading sources of mortality, with more than 700,000 annual deaths [1]. According to a recent WHO report, in 2019 1.3% of deaths were suicides. Although Confucianism remains influential in some parts of Asia, few studies provide insights into suicide from the perspective of Confucian culture. Confucianism is a civil religion integrating personal values, diffused broadly across the social fabric, resulting in relatively high levels of societal integration and lower suicide rates. Confucianism places value on male role obligations and devalues female roles within families and other social contexts, which may result in fewer male suicides and more female suicides. This study explores whether societies, where Confucian values are diffused across social arenas, have less suicide and lower suicide gender ratios compared with European and other societies, in part, due to religious and cultural influences. Confucianism should be explored in studies of suicide in Asian societies, both in theoretical and empirical studies.

### 1.1. Literature Review

Past studies suggest that economics, religion, and culture are important in explaining suicide variations. First, we review studies of the economic dimensions and influences on suicide, then discuss studies of religious and cultural effects on suicide.

### 1.2. Economic Factors Influencing Suicide

Platt [2] found suicide rates were likely to increase in times of economic depression and rapid social change. Durkheim demonstrated that economic environments could weaken social integration and regulation, resulting in higher suicide rates [3]. According to Durkheim, both economic growth and decline may increase suicides. Although they shared some features in common, some tests of Durkheim’s theory of social integration and regulation did not agree. Among them, Ginsberg [4] and Pierce [5] confirmed Durkheim’s claim that economic changes are the causes of suicidal behavior. However, Ginsberg [6] implicated that suicides declined during economic retraction and increased during economic expansion. Marshall and Hodge [6] implied that economic disruption was not as relevant to the suicide rates as economic hardship, deterioration, or improvement. Other empirical studies supported claims that economic deterioration led to an increase in suicides [7,8,9].

### 1.3. Religion and Suicide

Religious integration, identity, and beliefs are also related to suicide. Durkheim [3] used integration theory to explain how religious identity and belonging affect suicide rates. Fewer Catholics died by suicide compared to Protestants, when there was no difference between Catholic teachings regarding suicide. Rather, Catholics and Protestants varied in levels of social integration. Compared to Protestants, Catholics had a higher integration of beliefs, which may deter suicidal behavior.

Sociologists developed new theories that focused more on general influences on suicide, including the emphasis on variations in religious commitment. The theory of religious commitment and suicide [10] included eight principal beliefs focused on promises of bliss in the afterlife, suffering as the will of God, knowledge of God concerning human suffering, a heavenly system that is a fairer alternative to materialistic stratification, and confidence that God will assist in coping with hardships. Integrated religious beliefs become potential resources for enduring worldly sufferings instead of resorting to suicide.

In northern and eastern Europe, across Anglophone societies, in Latin America [11], and some Muslim countries, religious belief may insulate against suicide attempts, relieving mental and emotional stress. In Iran, intrinsic Islamic beliefs, even with high mental disorder scores, may result in fewer completed suicides [12]. In a nationally representative UK sample, Jacob, et al. [13] reported a negative correlation between religiosity and suicidality. However, the relationship between religion and suicide was positive for some groups. Compared to Unitarian/Universalist sexual minorities, unspecified Christian and Catholic sexual minorities had a higher risk of suicidal ideation [14]. In the prison context, although religiosity levels were high, they were not associated with suicidal ideation [15]. In China, among patients who received methadone maintenance therapy, Buddhist adherents had a significantly higher risk of suicide [16]. In contemporary China, young believers, a marginalized population, might suffer greater psychological strains, leading to higher suicide risk [17].

### 1.4. Culture and Suicide

Within certain contexts, secular culture affects suicidal behavior. Durkheim constructed a typology of four distinct suicide classes [3], in which societal integration and aspiration-permission were the main factors influencing suicidal actions. Egoistic and altruistic suicides reflected lower levels of societal integration. Anomic and fatalistic suicides were related to the aspiration-permission dimension. Durkheim [3] found that “suicide varies inversely with the degree of integration of social groups of which the individual forms a part”. Anomic and fatalistic suicides were opposing types with respect to social regulation. Anomie occurs when social norms are less compulsory, while fatalistic suicide arises from excessively oppressive regulations [18]. Although they do not easily apply to studies on culture and suicide [19], a number of theories, including the cognitive model of suicidal behavior [20], the interpersonal theory of suicide [21], and psychodynamic theories of desire to escape psychological pain [22] have been used to explain suicide [20].

Culture has variable effects on suicide. Empirical studies have focused directly or indirectly on cultural aspects of suicide across ethnic identities, religions, and societies. Suicide among Canadian aboriginal peoples has been reported in empirical studies. In British Columbia, suicides were lower for Aboriginal communities where native languages were still used [23]. Suicidal behaviors tended to rise [24] when aboriginal languages declined, which might lead to the disappearance of distinctive cultural identities. Among Hispanics in the United States, the disintegration of cultural values, such as familism and religiosity, might lead to acculturative stress, which may increase suicide risk [25]. Blacks in the United States who experienced greater acculturative stress, including feelings of social isolation, were more likely to commit suicide [26,27,28]. Conversely, culture could act as a protective factor. Latinos had lower risks of suicidal behaviors, possibly protected by culturally guided beliefs and support [29].

### 1.5. Confucianism and Suicide

Confucianism is an essential part of Chinese culture and remains a strong influence in Eastern Asia. According to Confucius and Mencius, although the preservation of biological life is good, individuals may sacrifice their lives to pursue the cardinal values of *ren* and *yi* (benevolence and justice). Additionally, Confucian belief emphasizes dignity to distinguish which people should actively end life [30]. Antithetical views of life in traditional Chinese thought are equally influential, summarized as follows: (a) despite the negative effects of suffering, achieving vocational purpose transcends killing self for dignity; (b) a broader scope of life commitment, rather than sacrificing life for limited or selfish causes [30]. Dimensions of Confucian ethics affecting thoughts on life and suicide include filial piety (*xiao*), harmony (*he*), and female gender role expectations (*nv xing jue se*) [31,32,33].

Research on Confucianism and suicide has made varying claims. Some scholars believed this traditional culture protected against suicide since suicide was regarded as the worst form of non-filial piety [34,35]. Alternatively, shared cultural beliefs, such as parental authority and the inferior position of females, rooted in Confucian beliefs, increased suicide risk among adolescents [32,36] and Chinese women [33,37]. Confucian beliefs may contribute to female suicide, reducing gender ratios compared with societies that are more individualistic and where regulative institutions may be weaker.

With respect to the relationship between suicide and Confucianism in China, a series of investigations have been conducted by the first author of this study and collaborating scholars. Zhang and Liu [38] found a positive relationship between Confucian ethics of female subordination and completed suicide [39], and found depression among female suicides in rural China. Jia and Zhang [33] concluded from an investigation among rural young Chinese with major depression that Confucian values appeared to be a protective factor for men but were a risk factor for women. Zhang, et al. [40] found most participants denied suicidal ideation directly and indirectly, viewing suicide as violating Confucian teachings.

Scholars called for attention to Confucian influences on suicidal behavior among Koreans. Affecting the Korean female Kim’s suicide attempt were Confucian values regulating family cohesion, parental authority norms, and child obedience [32]. Korean children who were expected by their parents to achieve academic success suffered from ‘education fever’ [36], resulting in academic stress, a suicidal behavior risk factor [41]. Traditional gender roles defined by Confucianism may affect suicide rates among Korean older male adults burdened with financial responsibilities [42]. Park, Baik, Kim and Lee [36] explained that cultural values can act as both a protective and destructive factor in suicidal behaviors. A qualitative study [35] conducted among Korean college students found Confucianism influenced reasons for not attempting suicide and reducing suicidal thoughts. Relying on Durkheim’s theory of suicide and the theory of cultural ambivalence by Wilkinson [43], Kang [44] concluded that conflicts between Western values of individualism and Confucian collectivism contributed to anomic and egoistic suicides.

Influenced by Confucianism, the Japanese place value on self-sacrifice as “a rule of the heart” [45], moral legitimacy, and positive evaluations of suicide [46]. Combined with animistic conceptions of deceased ancestral spirits, Confucian emphasis on loyalty, self-sacrifice, and honor contributes to altruistic suicide among Japanese youth [47]. Suicide prompted by social shame was acceptable in the Confucian context [48]. Iga [49] suggested that altruistic and fatalistic suicides motivated by shame were common in societies such as Japan and China, where social integration and social regulation were strong [48]. Prescribed ritual suicide by disembowelment was generally deemed as virtuous action against illegitimate authoritative directives, or as repentance for unforgivable sin [50].

### 1.6. Confucianism, Values, and Societal Integration

Confucianism may be understood as a religious expression, a sacred social ethics system, providing a shared sense of identity and transcendent values, integrated and diffused throughout society. Durkheim [51] defined religion as *collective representations* expressed through rites, including values and ideals, behavioral norms, and social role expectations. Geertz [52] refers to religion as a symbolic cultural symbol system of ultimacy and generality integrated across the social fabric. Confucianism is a social ethics system implying social control, connecting personal values to the broader culture and social order, firstly, as an authoritative system of sacralized values, and secondly, integrating social life across core social arenas and institutions, especially family, labor, community, and governmental realms.

Weber characterized Confucianism as a *status ethic*, reflecting traditional Chinese social order, represented as *world images* oriented toward governmental, military, and literary elites, but also familial obligations and roles [53]. These ideal norms locate transcendent values within a sacred cosmos, prescribing a legitimate, authoritative meaning system and order of social relationships. Weber [54] notes that for China, Confucianism divides religious labor with Buddhism, a counterpoint emphasizing contemplation and world rejection, reflecting traditional caste differences originating in feudal Asian societies.

Confucianism is a form of civil religion, a sacralized system of social ethics diffused across social settings and institutions, not institutionally differentiated, as with modern western Christianity. Bellah [55] defines civil religion as the “transcendent universal religion of the nation”, integrating sacred and ethnic identities and contributing toward social solidarity. Later, Bellah [56] applied the civil religion concept to twentieth-century Japan and China, expressed as national identity, values and aspirations. Civil religion in China and other Asian societies is diffused, permeating government, family, community social networks, and other institutions of civic life. Bellah’s analysis of civil religion in Imperial Japan and the Republic of China may be narrowly focused on personal values, minimizing the role of Confucianism’s influence on Asian national identity and institutions [57].

Strain theories emphasize the role of hope for the future in deterring violations of social norms [58,59]. Religious narratives center on theodicies, explanations of suffering, and human limitations [60]. Confucian values and role prescriptions are essentially civil religious, diffuse, and focused on national unity and identity. Confucian social ethics provide idealized images of social order centered around aspirations as applied to moral values and social relationships. Aspirations reinforce moral codes by increasing conformity with social norms and enhancing moral communities, contributing toward social cohesion and integration. Confucian values may have variable effects on suicide, especially for gendered role expectations regarding increased male familial obligations and devalued female status, as compared with societies and regions where societal integration around Confucian or other religious influences is weaker, and where psychological strain around role expectations is reduced.

This study tests two hypotheses: (1) Confucian countries have lower suicide rates than European countries, and (2) Confucian countries have lower suicide gender ratios than European countries.

## 2. Methods

### 2.1. Data Source

Data for this study came from the WHO Global Health Estimates from 2000 to 2019 [1]. Suicide rates were age-standardized according to WHO World Standard Population data for ease of comparison across countries, resulting in available data for 183 countries. Dependent variables included age-standardized suicide rates (all ages and both sexes) and male-to-female ratios for age-standardized suicide rates. Independent variables included WHO region, income level, categorized into World Bank categories of HI (high-income), UMI (upper-middle-income), LMI (lower-middle-income group), and LI (low-income), cultural influence (Confucianism and non-Confucianism). Although transformed over time, Confucianism is still the substance of learning, the source of values, and the social code of the Chinese. Its influence has also extended to other countries, particularly Korea, Japan, and Vietnam [61]. We included five nations where Confucian beliefs and teachings remain strong: China, South Korea, Japan, Vietnam, and Singapore. Classification of these five nations as Confucian societies has been discussed and vetted in previous studies [36,62,63,64]. What is more, suicide data issued by the Institute for Health Metrics and Evaluation [65] were employed as a reference to validate current data from WHO and make the study more convincing.

### 2.2. Statistical Analysis Plan

We applied one-way bivariate ANOVA analyses to examine potential differences in suicide rates and male-to-female ratio of suicides in terms of region, income level, and cultural beliefs.

ANOVA post hoc tests were conducted to specify potential differences between independent variable categories. After suicide rates and (male: female) ratios were found to vary by region, post hoc tests specified categorical differences.

Since no differences in suicide rates and gender ratios were detected, one-way ANOVA compared the five Confucian nations with 55 European nations to test for differences in suicide rates and suicide gender ratios across cultural contexts.

Multiple linear regression analysis identified factors of variables including suicide rates and gender ratios, controlling for region, income level, and religion-cultural context. After Confucian cultural status was determined to influence suicide rate gender ratios, one-way ANOVA analyses explored other factors which may affect suicide rate gender differences.

## 3. Results

Regional suicide rates and gender ratios were calculated from the 2019 WHO data. Suicide rates varied by region (*p* < 0.001). Post-hoc tests specify specific regional pairings which differ. Suicide rates in African countries were higher than that in the American region (*p* = 0.001), eastern Mediterranean region (*p* < 0.001), European region (*p* = 0.005), and south-east Asian countries (*p* = 0.007). There was no significant suicide rate difference between the African and western Pacific regions (*p* = 0.256). The suicide rate in the American region did not differ significantly from eastern Mediterranean countries (*p* = 0.376), the European region (*p* = 0.339), south-east Asia (*p* = 0.723), and western Pacific region (*p* = 0.075). No differences in suicide rates were found between the eastern Mediterranean and American regions (*p* = 0.376), European (*p* = 0.077), or south-east Asia region (*p* = 0.739). The suicide rate did not significantly vary between the European and other regions, except for the African region (*p* = 0.005). The same was true for suicide rates in South-East Asia in comparison with other regions. Suicide rates varied significantly between the western Pacific region and the eastern Mediterranean region (*p* = 0.017). A comparison of gender ratios for suicide rates revealed no significant regional differences (*p* = 0.918).

The results of the one-way ANOVA showed that there were no differences in suicide rates across countries in terms of income level. Differences in gender ratio existed across countries by income level (*p* = 0.031). Lower-income countries had the lowest gender ratio among all countries across income levels (*p* = 0.032, 0.004, 0.002 separately). There were no significant gender ratio differences between low-middle income, upper-middle-income, and higher-income countries.

According to Table 1, it can be concluded that there were no statistically significant differences in suicide rates and gender ratios between Confucian nations and non-Confucian nations (*p* = 0.736, 0.064, respectively). Comparisons between Confucian nations and European nations showed no statistical difference in suicide rates (*p* = 0.397), although the difference in gender ratio was significant (*p* = 0.008) (Table 2).

To examine key factors influencing suicide rates in Confucian and European countries, we used linear regression by setting the world region as dummy variable categories. Region, income level, and Confucian country status were also included. As shown in Table 3, income level and Confucianism were entered into the equation. ANOVA for the regression model was not significant (*p* = 0.616), indicating that none of the predictor variables significantly influenced suicide rates for Confucian and European countries.

To examine factors influencing the gender ratio of suicide rates for Confucian and European countries, we conducted another linear regression by setting world regions as the dummy variable. All three variables including world region, income level, and Confucianism were included in this model. As shown in Table 4, income level and Confucianism were significant forces for this model, as the ANOVA results were *p* = 0.027 < 0.05, indicating that at least one of the variables was a factor influencing the gender ratio in Confucian and European countries. According to t-test results, Confucianism affected the gender ratio of suicide rates in Confucian and European countries (*p* = 0.008).

Since gender differences in suicide rates occurred between Confucian and European countries, one-way ANOVA was used to compare gender suicide gaps for females and males, respectively. The results in Table 5 show that the suicide discrepancy for females (*p* = 0.003) was statistically significant, but not for males.

## 4. Discussion

In this study, we examined differences in suicide rates and gender ratio of suicide rates across 183 countries in terms of region, income level, and cultural belief (Confucianism), based on the rearranged 2019 WHO suicide data. Generally speaking, African countries had the highest suicide rates among all the countries. Among 47 African countries, 36 (76.6%) had suicide rates higher than the global average (9.0/100,000) [1]. A problem worth noting was that data quality from the majority of African countries was low, and might produce inaccurate suicide and suicide rate gender ratios. Concerning gender ratios, there were no overall differences among regions. One-way ANOVA showed that there were no discrepancies in suicide rates by income levels, although income influenced suicide rate gender ratios. Lower gaps between male and female suicides occurred in the lowest-income regions.

With respect to our hypotheses, we found no suicide rate gaps between Confucian nations and non-Confucian nations overall, whereas there were discrepancies in suicide gender ratios. The gender ratio was lower in Confucian compared with non-Confucian countries. Furthermore, both one-way ANOVA and multiple regression analysis suggested that Confucianism was the most powerful variable affecting differences in the 2019 suicide gender ratio between the five Confucian nations and European countries. Particularly, those nations where Confucianism remained influential had lower gender ratios than European nations, historically derived from Western culture and religion. This study confirmed our second hypothesis that Confucianism countries have suicide gender ratios lower than those of European countries. Our first hypothesis, which assumed lower suicide rates for Confucian societies, was not supported. In particular, the female suicide rate was relatively higher in the Confucian context than in Western countries, resulting in a lower suicide rate gender gap. As Standish [66] notes, Confucian-influenced societies are the only major world cultural group worldwide with higher suicide rates for females. Even Japan and South Korea which were among the nations with the highest suicide rates, female suicide rates (6.9/100,000 and 13.4/100,000 separately) were relatively higher than other countries. Consequently, the higher female suicide rates resulted in a smaller suicide gender ratio. The gender ratio finding is consistent with existing research conducted in China [38,67] and in the United States among adolescents [68].

The core beliefs of Confucianism support harmonious interpersonal relationships, including family cohesion, social integration, and collectivism over individualism [31]. Particularly, The *Three Cardinal Guides* [69] set an order that “the ruler guides his subjects, father guides son, and husband guides wife” in which the ethic of male power and authority was explicitly expressed. These ethics kept emerging in the Confucian classics and the inferior status of women was explicitly embodied in the highest ethical requirements as expressed in *The Three Obediences* [70], which demanded female obedience to men. In childhood, females obeyed fathers and brothers, after marriage they obeyed husbands, and after spousal death they obeyed sons [70]. Under traditional values of male superiority, women were deprived of independent personality or other expressions of executive agency, and were expected to make more sacrifices [38,71] while being less valued than males [72]. Norms of female inequality were formally legalized in Confucian societies. Suppression of women remains in contemporary Asian societies, such as China [73], South Korea [74], Vietnam [75], and Japan [76]. In Confucianism-based patriarchal cultures, female role expectations are shaped by the ideology of moral women restricted to domestic contexts, requiring women to make great sacrifices for their families [77]. For example, being a caring mother has been at the center of Confucian values. Marriage has been a fundamental expectation for women [78]. In contemporary Confucian societies, aspirational Western values of gender equality and female autonomy have been gaining credibility. Some women reject the bondage implied in traditional Confucian gender role expectations, perceived as oppressive and restricting life choices. In this way, two conflicting forces are fostered, contributing to psychological strain [58]. To link psychological strain and suicidal attempts, Zhang [58] developed the strain theory of suicide, which has been tested in various settings with varied samples and has been validated as a key predictor of suicidality [79]. According to the strain theory of suicide, when females perceive value strains generated by tension between traditional and modern beliefs, suicidal ideation and attempts increase [38,39].

In addition to value tensions, Confucian emphasis on filial piety may have an important role in insulating men from suicide. From the traditional Confucian perspective, the relationship between parents and sons is viewed as the foundation of morality. Filial piety is particularly stressed in Confucian society within this context [78]. Compared with women, men are assigned more responsibility to fulfill expectations of filial piety [78,80], including absolute submission of the next generation to the previous generation, care for physical bodies as they are given by parents, and expectations that sons will outlive parents. Since filial piety is regarded not only as an obligation of sons but also as a core moral principle [78], men feel more obligated not to die of suicide so as not to dishonor their parents and their communities [81]. Compared with males, Confucianism does not act as a protective factor for females against depression and suicide attempts. The positive association between the Confucian ethic of female subordination and mental issues, including depression and successful suicide, has been affirmed in studies in mainland China [33,38,39,67].

Although culture cannot explain every aspect of suicide, this current paper reconfirms that cultural differences contribute to gender differences in suicide in some societies. In other words, results derived from 2019 WHO data provide new evidence from a broader sample, including countries beyond mainland China, for the influence of Confucian beliefs on suicidal behavior among Asian females. Therefore, a lower male-to-female gender ratio is expressed in relatively high female suicide rates in Confucian countries.

This study has certain limitations. The 2019 WHO suicide data did not have high overall quality, in which just over sixty countries (32.8%) had high-quality vital statistics registration [1]. In addition, among the 55 European and Confucian nations, 18 nations did not provide national death registration data with high completeness and quality. These issues might reflect bias leading to unreliable conclusions about the relationship between Confucianism and suicide. What is more important, current data from WHO did not comprise comprehensive information concerning suicide that might be able to explain current results from different perspectives. Future studies in this area of Confucianism and suicidality may benefit from more high-quality data and the inclusion of more variables about included nations. Helpful variables might include variations in religious commitment and values, secular and other cultural values and resources for female independence, as well as time series data to track the impact of economic, childbearing, and other personal and shared events that may influence temporal and spatial suicide variations.

## 5. Conclusions

Confucianism is characterized by filial piety (respecting and abiding parents and elderly in the family) and subordinate female status, especially within family contexts. Confucian beliefs remain strong and influential in some Asian countries, imposing adverse effects on females that result in relatively higher suicide risks for women. While the suicide rates for men are comparatively low, the rates for women are relatively high, and the narrowed discrepancy contributes to lower suicide rate gender differences.

## Figures and Tables

**Table 1 ijerph-20-02188-t001:** Summary of the suicide rates and the gender ratios in the world by Confucian belief.

Confucian Belief	N	Suicide Rates		*F*	*p*	Male: Female Ratio		*F*	*p*
		Mean	SD			Mean	SD		
Confucian nations	5	11.4	5.9	0.114	0.736	2.1	0.3	3.466	0.064
Non-Confucian nations	178	10.0	8.8			3.7	1.8		
Total	183	10.1	8.7	-	-	3.6	1.8	-	-

**Table 2 ijerph-20-02188-t002:** Comparison between the Confucian and European countries in the suicide rates and the gender ratios.

Confucian Belief	N	Suicide Rates		*F*	*p*	Male: Female Ratio		*F*	*p*
		Mean	SD			Mean	SD		
Confucian nations	5	11.4	5.9	0.731	0.397	2.1	0.3	7.551	0.008
European nations	50	9.5	4.6			3.7	1.2		
Total	55	9.7	4.6	-	-	3.5	1.3	-	-

**Table 3 ijerph-20-02188-t003:** Major factors of suicide rates in Confucian and European countries.

Variables	B	Std. Error	Standardized Coefficients Beta	*t*	*p*	*F*	*p*
(Constant)	9.881	3.621		2.729	0.009	0.490	0.616
Income level	0.441	0.866	0.070	0.509	0.613		
Confucianism	−1.922	2.214	−0.119	−0.868	0.389		

**Table 4 ijerph-20-02188-t004:** Major factors of gender ratio of suicide rates in Confucian and European countries.

Variables	B	Std. Error	Standardized Coefficients Beta	*t*	*p*	*F*	*p*
(Constant)	2.514	0.913		2.755	0.008		
Income level	−0.113	0.218	−0.067	−0.517	0.607	3.857	0.027
Confucianism	1.533	0.558	0.356	2.747	0.008		

**Table 5 ijerph-20-02188-t005:** Comparison between the Confucian and European countries in the suicide rates of males and females.

Confucian Belief	N	Female Suicide Rates		*F*	*p*	Male Suicide Rates		*F*	*p*
		Mean	SD			Mean	SD		
Confucian nations	5	7.1	3.7	10.046	0.003	15.8	8.4	0.011	0.917
European nations	50	4.1	1.9			15.4	8.2		
Total	55	4.3	2.2	-	-	3.5	8.2	-	-

## Data Availability

Data available in a publicly accessible repository that does not issue DOIs.Publicly available datasets were analyzed in this study. This data was accessed on 15 January 2022.

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
