# Peer review of "Confucianism and Gender Ratios of Suicide in the World: A WHO Data-Based Study"

_ijerph, 2023, doi:10.3390/ijerph20032188_

Round 1
Reviewer 1 Report
The article is well written and significant for the study of the suicide phenomenon from a religious perspective. The importance and the relevance of the Confucianism in Asian sociaties is well developed and discussed. The methodology is adequate and ambitious, even if the limitation to suicide rates and gender ratio, together with the use of only one WHO report can be limitating for the readership.
It could be interesting if the authors analyse also the selected five countries separately, since the presence of Japan, for instance, one of the countries with the highest suicide rate, could be misleading for the presented data.
The title should be changed: not only gender ratio is discussed in the paper.
The prevalence of more women committing suicide compared to other countries in the world is very peculiar. Have the authors thought to also analyse the way chosen to commit suicide? Since the authors partially justify this data on how is the 'merit' of a more inclusive society for women, the suicide mode may also differ and be more like that of the population in their own country. Please, try to discuss also this aspect.
Author Response
Thanks a lot for the comments.
With regard to the comment of analyzing the five countries separately, our thoughts were as follows: first, we were supposed to analyze the suicidal phenomenon shared among the Confucian countries which we believed the Confucian beliefs might have worked as a mechanism; Second, it was worth being addressed that Japan and South Korea were both among one of the highest suicide countries. Nonetheless, we believed it was beyond our research purposes that were mentioned above.
Regarding the change of the tile, although we discussed suicide rates among different countries, it was basic information for further discussion of the disparity in the suicide gender ratio. It is inappropriate to change the title.
It was inspiring to highlight the way of suicide to explore the suicide pattern in different countries and we believe this will be beneficial for suicide prevention. However, this will not be consistent with our research purposes that mainly focused on the suicide gender ratio within the Confucian context.
Sincerely,
Wei Wang
Jie Zhang
Wayne L. Thompson

Reviewer 2 Report
This paper deals with how Confucianism affects suicide rates by gender. It's an interesting perspective for the study of the suicidal phenomenon in the society.
Paper is well written and discussed.
It's worthy of publication after minor revions:
- In the opinion of the Reviewer, authors should delve into the statistics for the countries examined individually as well in order to understand whether the increased suicide rate in general may influence the total figure.
Please, discuss this aspect.
Author Response
Thanks a lot for the comments.
We have made some revisions which could be tracked in the discussion.
Sincerely,
Wei Wang,
Jie Zhang,
Wayne L. Thompson

Round 2
Reviewer 1 Report
The requested revisions weren't performed.
Author Response
Dear reviewer,
Thank you very much for your comments. Here are our responses.
Point 1: The research design can be improved and this study should not just be limited to WHO data.
Response 1: We just revised this part by adding more information on suicide rates around the world which has been presented in the manuscript.
Point 2: English language and style are fine/minor spell check required.
Response 2: We have checked the grammar and spelling problems and made corrections. Both of the revisions were done under the Tracking-changes mode.
Thanks.
Sincerely,
Wei Wang
Jie Zhang
Wayne L. Thompson
